# Amino Acid Content in the Muscles of the Red Deer (*Cervus elaphus*) from Three Types of Feeding Grounds

**DOI:** 10.3390/ani14192763

**Published:** 2024-09-25

**Authors:** Anna Kasprzyk

**Affiliations:** Department of Animal Breeding and Agricultural Consulting, University of Life Sciences in Lublin, 13 Akademicka Street, 20-950 Lublin, Poland; anna.kasprzyk@up.lublin.pl

**Keywords:** protein, nutritional value, biological value, meat, game, venison

## Abstract

**Simple Summary:**

An increase in game meat consumption has been observed worldwide. This is the first study to assess the amino acid composition and the protein quality of the muscles of deer from organic feeding grounds. This study assessed the nutritional value of the protein of the muscles of deer originating from forest, organic, and conventional feeding grounds. The meat of deer from the forest and the conventional feeding grounds exhibited higher exogenous and endogenous amino acid contents than that of deer from the organic feeding ground. However, the meat of deer from the organic farm feeding ground is characterized by the highest protein content and ensures optimal satisfaction of the demand for essential amino acids. The highest lysine content was found in the muscles of the deer from the forest feeding ground. This knowledge can help nutritionists develop diets that meet the nutritional recommendations for different consumer groups and breeders to make decisions about farm location and deer feeding strategies.

**Abstract:**

This study aimed to analyze the amino acid profile, with a particular focus on the nutritional value of the protein of the *longissimus lumborum* (LL) and the *semimembranosus* (SM) muscles of deer originating from three feeding grounds: forest (FFG); conventional (CFG) grounds; organic farm (OFG). This is the first time that deer from an organic farm feeding ground have been included in this study. The muscles were collected from 36 deer carcasses with equal proportions of sex and 31 months of age. This study demonstrated significantly higher essential amino acid (EAA) and non-essential amino acid (NEAA) contents in the muscles of deer from the FFG and CFG compared to the OFG. However, the EAA-to-NEAA ratio was significantly higher for the muscles of deer from the OFG. The muscles of the FFG and CFG deer were characterized by a higher concentration of lysine as well as acidic and tasty amino acids compared to the OFG deer, with the muscles of the latter exhibiting a higher percentage of branched-chain amino acids (BCAA). The results obtained can be used professionally by nutrition specialists in preventive and therapeutic diets and breeders to make decisions about farm location and deer feeding strategies.

## 1. Introduction

A healthy lifestyle increases the demand for animal protein and meat with a reduced fat content. As a result, there is a growing interest in game meat, which, compared to livestock meat, is healthier [1]. The *Cervidae*, or the deer family, live in their natural habitat and feed on natural food, making use of abundant and rich forest vegetation and wasteland, where the primary forage includes grasses, herbs, leaves, tree bark, acorns, beech nuts, and berries [2]. Global game meat consumption in 2023 exceeded over 2 million tonnes [3]. In order to increase supply, different deer species are bred, e.g., in Europe, the USA, and New Zealand [4]. An increase in the consumption of both game and venison meat is particularly observed in Europe, and its consumption is now associated with prestige [5]. Poland is among the leading game suppliers in Europe. Over the past 20 years, Poland has observed a steady increase in the deer population, from 92,000 in 2001 to 293,000 in 2022 [6], of which 40,078 deer are kept in 900 Game Breeding Centres [7]. The development of Polish cervid breeding accelerated after Poland’s accession to the EU. Farmers searching for a new niche activity became interested in using poor agricultural land to breed the *Cervidae*. Many small, extensive, and organic farms, as well as breeding parks for this species, have, therefore, been established [8]. In Poland, from 5223 tonnes (2015) to 5692 tonnes (2023) of deer meat are harvested annually during hunting season [7]. In 2022, the consumption of game meat in Poland amounted to 0.084 kg/person [9]. Game meat is perceived as a luxury food [5]. It is very difficult to find in Polish retail shops, as it is an exclusive export product. 

To date, numerous studies have been published on the meat of free-living deer and those originating from conventional farms, affecting its basic chemical composition [9,10,11,12,13,14,15,16,17,18,19,20,21,22]. Proteins are the main components of the muscles; however, total protein content does not fully indicate the nutritional value of the meat. Therefore, it is necessary to determine the amino acid content of meat [23,24]. Of the 18 amino acids important in human nutrition, 8–10 (depending on age) cannot be synthesized by the human body and must, therefore, be supplied through the diet [25]. These include exogenous amino acids: isoleucine; leucine; lysine; methionine; phenylalanine; threonine; tryptophan; valine; as well as arginine and histidine in the case of infants [23,25]. Amino acids are used for the synthesis of systemic proteins. Animal proteins are responsible for the efficient functioning of the body, allowing it to grow and recover [25]. In addition, they regulate gene expression and are essential precursors for the synthesis of hormones [23]. Proteins regulate blood pressure as well as glucose and lipid metabolism [26]. Importantly, low protein intake, with equivalent amounts of animal and plant proteins provided, does not meet the requirement for exogenous or essential (EAA) amino acids. EAA deficiencies limit the regeneration and detoxification of the tissues and organs and contribute to the weakening of the immune and nervous systems [27].

Despite the publication of numerous studies, there is insufficient knowledge in the literature on the composition of amino acids in deer meat. Articles published by Lorenzo et al. [28], Okuskhanova et al. [29], Pérez-Serrano et al. [30], and Pérez-Serrano et al. [31] are among the few available articles concerning the composition of amino acids in the wild red deer muscle tissue. This study is the first to assess and analyze the amino acid profile, with a particular focus on the nutritional value of the protein of two commercially important primal cuts of raw game meat (the *longissimus lumborum* and the *musculus semimembranosus*) of the deer originating from the forest (FFG), organic farms (OFG), and conventional farm feeding grounds (CFG). These findings could have significant implications for deer meat producers and consumers.

## 2. Material and Methods

### 2.1. Animals and Muscle Sampling

The red deer originated from three types of feeding grounds: forest (FFG); organic farms (OFG); and conventional (CFG). The animals lived in the northeastern part of Podkarpackie Voivodeship in Poland. The forest feeding ground covers 8000 ha and is composed of considerable species richness of trees, shrubs, meadow plants, and shrublets [32]. The composition and nutritional value of vegetation consumed by the free-living red deer, due to the size of the area, was not controlled. Additional food for the red deer was also provided with agricultural crops. Animals from the OFG were raised at the farm with an organic farming certificate covering the meadows and pastures with ponds and watercourses, the animals, and the entire rearing process [33,34]. The animals were reared on the farm with a density of 0.20 LU/ha per pen and were kept in their natural habitat, respecting the EU Directive, 2010/63/EU. In the conventional system, the stock density of 0.52 LU/ha was applied according to the DEFRA [35] and FEDFA [8] recommendations. The feed for the animals was provided by a natural grazing ground.

Information on the vegetation characteristics and the nutritive value of the feeding grounds was described in detail previously [9]. Briefly, floristic analyses of the feeding ground were carried out once a month from April to October. In each feeding ground, 35 phytosociological relevés were made using the Braun–Blanquet method [36]. The OFG was characterized by the presence of 116 plant species, with 60 species exhibiting phytoncidal properties. The structure of the natural food resources for deer from the OFG was composed of 42% herbaceous dicotyledonous species, 18% legumes, 13% grasses, 12% shrubs and shrublets, 7% sedges and rushes, 5% deciduous trees, 2% coniferous trees, and 1% ferns. In turn, 78 plant species, including 48 phytoncide plants, were identified in the CFG. The food resources on the CFG were as follows: 36% herbaceous dicotyledonous species; 17% grasses; 12% shrubs and shrublets; 10% legumes; 10% deciduous trees; 9% sedges and rushes; 5% coniferous trees; and 1% ferns. Constant accessibility to water and multi-ingredient licks (SOLSEL^®^ Wild, Kassel, Germany) was provided on both farms. 

The experiment was conducted according to the guidelines of the Declaration of Helsinki and in compliance with the European Union law [37] (received in Poland by Legislative Decree 266/2015) of the European Parliament and the Council on the protection of animals used for scientific or educational purposes. No procedures on animals were performed that would require ethical protocols, and slaughter resulted from the production cycle on the farm and, according to Polish legislation [38], was treated as a routine agricultural activity. Red deer were shot in October, with the consent and supervision of veterinary services. Animals were approximately three years old, as estimated by tooth eruption. After shooting, animals were immediately bled out and then were transported (ca. 30 km; 1 h) under refrigerated conditions in a lorry into the game carcass handling unit, where they were eviscerated. Muscle samples (approx. 200 g) for testing were collected during routine carcass-cutting operations at a processing plant. Thirty-six samples of the longissimus lumborum (LL) muscle and 36 samples of the semimembranosus (SM) muscle (6 hinds and 6 stags in each feeding ground) at 48 h *post mortem* were taken for analysis. These samples were vacuum-packaged in polyethylene bags and stored at −20 °C in a chilling room until laboratory analysis. The experimental schema is shown in Table 1.

### 2.2. Chemical Determinations

Total protein content was determined using the Kjeldahl titration method (procedure 950.36; (N × 6.25) according to AOAC [39]. The determination of the amino acid contents, including aspartic acid, threonine, serine, glutamic acid, proline, glycine, alanine, valine, isoleucine, leucine, tyrosine, phenylalanine, histidine, lysine, and arginine was carried out in accordance with the AOAC method [39]. Lyophilized deer muscle tissue samples were hydrolyzed with 6 N HCL for 22 h at 110 °C in the presence of nitrogen. After evaporation of the hydrolysate, the amino acids were dissolved in citrate buffer (pH 2.2) and subjected to chromatographic analysis in an amino acid analyzer. Cystine, cysteine, and methionine of the protein were oxidized with performic acid to cysteic acid and methionine sulphonate and then hydrolyzed in 6 N hydrochloric acid at 110 °C for 18 h. After evaporation of the hydrolysate, the amino acids were dissolved in citrate buffer (pH 2.2). The solution prepared in this way was collected for chromatographic analysis. The tryptophan content was determined in a sample of the protein hydrolyzed in a barium hydroxide solution at 110 °C for 18 h. Barium ions in the hydrolysate were precipitated with sulphuric acid. The precipitate was centrifuged, and the solution was transferred to a volumetric flask. The precipitate was rinsed with citrate buffer (pH = 2.2), with the solution being transferred each time to the volumetric flask [40]. The solution prepared in this way was collected for chromatographic analysis. The chromatographic analysis was performed in an AAA 400 amino acid analyzer (INGOS, Prague, Czechia) with an ion-exchange column and a UV-VIS detector. Each sample was analyzed in triplicate. The results were expressed in mg/g of the muscle tissue.

### 2.3. Protein Nutritional Value Assessment Parameters and Biological Value

The following were calculated: total amino acid (TAA) content; essential amino acid or exogenous amino acid (EAA); non-essential amino acid or endogenous (NEAA); the percentage of essential amino acids (%EAA); non-essential amino acids (NEAA), namely, threonine (Thr), valine (Val), methionine (Met), isoleucine (Ile), leucine (Leu), phenylalanine (Phe), lysine (Lys), tryptophan (Trp), histidine (His); and the percentage of endogenous amino acids (%NEAA), i.e., arginine (Arg), serine (Ser), aspartic acid (Asp), glutamic acid (Glu), proline (Pro), glycine (Gly), alanine (Ala), cysteine (Cys), tyrosine (Tyr), the exo-to-endo ratio = EAA/NEAA, Met + Cys, acidic amino acids (AAk) = Asp, Glu, aromatic amino acids (AAa) = Phe, Tyr, Trp, tasty amino acids (DAA) = Asp, Glu, Gly, Ala, branched-chain amino acids (BCAA) = Val, Ile, Leu, large neutral amino acids (LNAA = Val, Ile, Leu, Phe, Tyr), the EAA/TAA ratio, and the index F = BCAA:AAa. 

Based on the amino acid content of the muscles, the chemical score (CS) and the essential amino acid index (EAAI) were calculated. The chemical measure of protein quality CS was determined as a ratio of the exogenous amino acid content of the test protein (g/100 g of digestible protein) to the content of the same amino acid in the standard protein (g/100 g of protein) [41]. For the calculation of CS1, CS2, CS3, EAAI1, EAAI2, and EAAI3, three standards were used, i.e., amino acid standards for an adult from 1991 [42], 2002 [43], and 2013 [44], respectively (Table 2). 

EAAI, similarly to CS, is based on the comparison of the composition of exogenous amino acids of the protein under assessment and standard protein, but it also takes into account the presence of all amino acids essential for the synthesis of proteins, and was calculated according to the following formula:

EAAI (%) = *n*^log EAA

where log EAA = [1/*n*] × [log (100 a1/a1R) + … + log (100 a *n*/a *n* R)];

*a*—mg of amino acid in 1 g of studied protein;

*aR*—mg of amino acid in 1 g of reference protein;

*n*—the number of amino acids taken into account in calculations (the methionine-cysteine pair counts as 1).

The expected value of the protein efficiency ratio (PER) was calculated using the three regression equations provided by Alsmeyer et al. [45]:

PER1 = −0.468 + 0.454 × Leu − 0.104 × Tyr;

PER2= −1.816 + 0.435 × Met + 0.780 × Leu + 0.211 × His − 0.944 × Tyr;

PER3 = 0.08084 × (X7) − 0.1094,

where X7 = Thr + Val + Met + Ile + Leu + Phe + Lys.

The protein biological value was calculated according to Oser [46], taking into account the EAAI:

Biological value (BV) = 1.09 (EAAI) − 11.7.

### 2.4. Statistical Analysis 

Statistical analysis of the data obtained was performed using Statistica 13 (TIBCO Software Inc., Palo Alto, CA, USA). The normality of the data distribution was tested using the Shapiro–Wilk test. The effects of feeding ground and sex were estimated using a two-way ANOVA model, including interaction. Since the sex and interaction were never significant, they were eliminated from the model. In order to compare the individual mean values, Duncan’s post hoc test was conducted. Differences were considered significant at *p* < 0.05. The tables illustrate the average values, the standard error of the average value, and the significance levels. 

## 3. Results

Protein is one of the most important nutrients. Ensuring adequate protein intake of high-quality protein is crucial for human health. The results provided in Table 3 indicate a significantly lower protein content of the LL and SM muscles of the FFG deer than that of the OFG deer and the CFG deer.

The content and composition of amino acids in meat are important elements in assessing the nutritional value of raw material. This study presented, for the first time, the results of the amino acid content in the muscles of the deer from the organic feeding ground. The feeding ground was found to have a significant effect on the content of most amino acids. This study noted a significantly higher content of exogenous (EAA) and endogenous or non-essential (NEAA) amino acids in the LL muscle of the FFG deer and the CFG deer, as compared to the OFG deer. This was mainly due to the higher content of the following amino acids: Thr; Val; Ile; Lue; Phe; Lys; Glu; Asp; Ala; Ser; Tyr. The highest Asp content was determined in the LL muscle of the FFG deer, whereas the lowest was noted for the OFG deer. The highest Phe + Tyr content and percentage of endogenous or non-essential (NEAA) amino acids were exhibited by the LL muscle of the FFG deer, whereas the lowest values were noted for the OFG deer (*p* ≤ 05).

The LL muscle of the FFG deer and the CFG deer, compared to the OFG deer, was characterized by a significantly higher percentage content of acidic and tasty amino acids. Acidic amino acids accounted for approximately 27% of the pool of all amino acids. A significantly higher and more favorable EAA:NEAA ratio was noted for the LL muscle of the OFG compared to the other two groups (FFG and CFG). Regarding the percentage content of branched-chain amino acids (BCAA), the LL muscle of the OFG deer exhibited the highest level. As far as the tasty amino acid (DAA) content is concerned, significantly higher levels in the LL muscles of the FFG deer and the CFG deer were noted. The LL muscle of the OFG deer was characterized by a higher percentage of the large neutral amino acids (LNAA) and a significantly lower DAA:TAA ratio than that for the deer from the forest feeding ground.

The assessment of the nutritional value of the protein was carried out based on the amino acids contained in the muscles compared to the requirement for an adult. To this end, a chemical score (CS) was used, which enabled the rapid and simple determination of the quality of the test protein by comparing its amino acid composition to the composition of the protein taken as a standard and the indication of the limiting amino acid. A limiting amino acid index CS equal to 84 denotes a level below which the amino acid of a particular protein is characterized by a biological value that is unsatisfactory for normal body development. The chemical score, the efficiency coefficient, and the biological value of the protein of the LL muscle of the deer are provided in Table 4. All the calculated average chemical scores CS for three protein standards (FAO/WHO, 1991; USA, 2002; FAO, 2013) amounted to over 100. A particularly high value of the CS index was noted for Trp, Thr, Phe + Tyr, and Ile. The highest CS 1-3 values for Thr and Phe + Tyr and, at the same time, the lowest CS 1-3 value for Met + Cys were exhibited by the LL muscle of the FFG deer. The CS 1-3 index for Ile was higher for the LL muscle of the FFG deer and lower for the OFG deer, with an intermediate value noted for the CFG deer. There was a significantly higher value of the CS1-3 index for valine in the LL muscle of the FFG deer compared to the other groups. The calculated PER and BV indices confirm the high nutritional value of the protein of the LL muscle. A significantly higher value of the PER3 index was exhibited by the LL muscle of the FFG deer and the CFG deer. As for the biological value (BV) of the protein, no significant differences were noted in the LL muscle of the analyzed deer groups. 

After analyzing the SM muscle (Table 5), a significantly higher protein content was observed in the OFG deer and the CFG deer compared to the FFG deer. The muscles of the OFG deer were characterized by significantly lower Thr, Ser, Asp, Glu, and Tyr contents compared to the muscles of the FFG deer and the CFG deer. The highest Met (*p* ≤ 0.017) content was noted for the SM muscle of the CFG deer, whereas the lowest was for the FFG deer. Regarding Leu, its significantly higher content was exhibited by the SM muscle of the CFG deer compared to that of the OFG deer. Protein of the SM muscle of the deer from the forest feeding ground was found to be a rich source of Asp and to exhibit the highest content of this amino acid (*p* ≤ 0.010). The SM muscle of the CFG deer was characterized by intermediate values for Phe, Gly, and Ala compared to the two other groups. 

In general, the SM muscle of the OFG deer showed lower EAA and NEAA contents compared to the muscle of the FFG deer and the CFG deer. An analysis of variance showed significant differences in the EAA:NEAA ratios between the OFG deer and the FFG and CFG deer. A significantly higher percentage of the acidic amino acids and Phe + Tyr was shown by the SM muscles of the FFG deer and the CFG deer. The muscles of OFG deer, compared to the CFG deer, were characterized by a higher percentage of BCAA and LNAA contents. A significantly higher DAA content was noted for the meat of the FFG deer and the CFG deer. The highest percentage of Met + Cys content (*p* ≤ 0.023) was exhibited by the SM muscle of the CFG deer and the lowest by the FFG deer. The DAA:TAA ratio was significantly higher for the muscles of the FFG deer and the CFG deer compared to the OFG deer. The highest value of the CS 3 index (Table 6), in relation to Lys, Thr, and Phe + Tyr, was exhibited by the SM muscle of the FFG deer and the lowest by the OFG deer. Significant differences between the SM muscle of the CFG deer and that of the other groups (FFG and OFG) were only noted for the PER 2 index. The SM muscles of the OFG deer were characterized by a significantly lower percentage content of EAAI compared to the FFG and CFG groups. The SM muscles of the deer from the conventional feeding ground exhibited a higher biological value of the protein than the muscles of the deer from the forest and organic feeding grounds.

## 4. Discussion

The nutritional value of meat products is primarily determined by the protein content. Assessment of this component is essential to determine its suitability for the human diet, as dietary protein is an important source of amino acids for the synthesis of systemic proteins [25]. In general, the protein content of the deer meat was high, and in the LL muscle, it ranged from 21.42% (FFG) to 23.29% (OFG), whereas in the SM muscle, it ranged from 22.25% (FFG) to 23.80% (OFG). Similar results for the protein in the *longissimus thoracis* and *lumborum* muscles for wild deer trapped in the autumn and winter were reported by Pérez-Serrano et al. [30] and Ugarković et al. [18] for the wild axis deer. Wild deer from Spain and New Zealand exhibited the following values of this index: 22.7% and 24.1%, respectively [31]. The protein content of the meat of springboks originating from four production regions was lower and ranged from 18.80% to 21.16% [47]. According to the EFSA recommendations, protein intake for adults should be 0.83 g/kg of body weight on a daily basis [48]. More recent studies suggest that the recommended daily allowance (RDA) for protein for the elderly, athletes, children, and adolescents should be higher, at a level of 1.1–1.2 g/kg of body weight [44]. Thus, the intake of 215 to 311 g of deer meat entirely satisfies an adult’s protein requirements.

However, complete protein content significantly affects and determines the value of the meat, as the protein quality is determined by the amino acids it contains. This study shows that deer meat is a source of protein with a high biological value. It contains all of the eight amino acids essential for adults and all the nine amino acids essential for children. The LL and SM muscles of the OFG deer showed a lower exogenous (EAA) and endogenous or non-essential (NEAA) amino acid content compared to the muscles of the CFG deer and the FFG deer. According to Pérez-Serrano et al. [31], the amino acid profile of deer meat is linked to the diet. The main nutritional factors that affect protein deposition include the amount of energy supply and the provision of protein of adequate quantity and quality in a food ration [49]. Bulky feeds are the main source of food for ruminants and account for approximately 83% of the feed ration for beef cattle [50]. Metabolized protein in ruminants originates from microbial protein synthesized in the rumen and from protein that does not undergo microbial fermentation [51]. Microbial protein entering the small intestine is the main form of nitrogen used by ruminants and is a high-quality source of amino acids, which are then deposited in the tissues. The diet of ruminants has a significant impact on the microbial population, affecting the relative quantity of microorganisms. In addition, it was reported that interactions between bacteria and protozoa may also affect the synthesis of microbial proteins [52,53]. It is also known that an increase in the sulfur amino acid content increases the provision of essential dietary amino acids to ruminants [54]. Polyphenol oxidase (PPO) protects plant proteins against degradation in the rumen. The higher amino acid content observed in this study in the muscles of the CFG deer, compared to the OFG deer, can be explained by the fact that in the feeding grounds, the predominant vegetation (almost 60% vs. 46%) included grasses and green dicotyledonous plants, e.g., the red clover (*Trifolium pratense*) and the cock’s-foot (*Dactylus glomerata*), which are a rich source of PPO [54,55]. The lower amino acid content of the muscles of the OFG deer analyzed in this study may be due to the varied diet and is possibly linked to extensive AA degradation in the rumen, resulting in a lower amount of amino acids absorbed in the small intestine [56]. However, the LL and SM muscles of the deer from the organic feeding ground (OFG) were characterized by a more favorable EAA:NEAA ratio than that for the two other groups (0.84 and 0.87 vs. 0.79 and 0.81 for the FFG deer; 0.82 and 0.83 for the CFG deer). It follows from the above that these ratios in the protein of the muscles of the deer from the organic feeding ground were more desirable than those in the protein of the FFG deer and the CFG deer. These differences were probably caused by a variety of physiological mechanisms. The lack of information on the amino acid composition of the muscles of the organically farmed deer in the available literature makes it difficult to discuss this topic. It is noteworthy that the muscles of the deer from the organic feeding ground showed a higher percentage of BCAA amino acids, which appears beneficial in light of studies confirming the role of these amino acids in performing many biochemical functions in the brain [57], lowering cholesterol levels, and protecting the liver [58]. In addition, they accelerate anabolism of muscle mass, increase muscle endurance, and are commonly used in nutritional supplements for athletes [59].

The amino acids present at the highest levels in the present study included glutamic acid (an average content of 3120 mg/100 g in the LL muscle) and lysine (1710 mg/100 g), followed by aspartic acid (1680 mg/100 g) and leucine (740 mg/100 g). The profile obtained in this study is consistent with the profile reported for the wild red deer [28]. Similar values for Glu and Lys were reported by Pérez-Serrano et al. [30], who analyzed the *longissimus thoracis* and *lumborum* muscles of the Iberian wild red deer. Higher values for the above-mentioned amino acids were observed by Ugarković et al. [18] for the *longissimus thoracis* muscle of the wild axis deer. According to Hoffman et al. [47], the two amino acids present at the highest levels in the *M. longissimus dorsi (LD) muscle of the springbok* included glutamic acid and aspartic acid, with the observed values ranging from 2470 to 2740 mg/100 g, and from 2310 to 2540 mg/100 g, respectively. The two most abundant exogenous amino acids are lysine and leucine, whereas methionine was present at the lowest levels [18,28,30], which was also confirmed by the results of the current study. On the other hand, glutamic acid, aspartic acid, and alanine were most abundant in the endogenous amino acid fraction, as in the studies by Lorenzo et al. [28], Pérez-Serrano et al. [30], and Pérez-Serrano et al. [31]. For endogenous amino acids, the lowest values in the authors’ own study were observed for cysteine (53–102 mg/100 g). In turn, Lorenzo et al. [28] observed the lowest values for tyrosine (714–742 mg/100 g) and proline (776–798 mg/100 g). The EAA:NAA ratio obtained in that study was similar to the values noted for the Iberian wild red deer by Pérez-Serrano et al. [30] and for the wild axis deer by Ugarković et al. [18]. Slightly higher values for the wild deer were reported by Lorenzo et al. [28], whereas considerably higher values for deer from Spain and New Zealand were reported by Pérez-Serrano et al. [31].

The least amount of exogenous amino acids was contained in the LL and SM muscles of deer from the organic feeding ground, as 100 g of the tissue contained 8.14 and 8.02 g of these amino acids, respectively. The daily requirement for EAA for an adult is 5.59 g [60]. Therefore, 100 g of the muscle tissue covers the daily requirement for this component in excess. Amino acids perform important physiological roles in the body and may also have specific health-promoting effects [61]. Therefore, the high lysine and arginine content of deer muscles reported in the present study is noteworthy. Lysine, which is abundant in the meat of the deer from the three feeding grounds, is essential for proper bone formation and growth of children [13,62]. It is involved in the synthesis of antibodies and hormones, contributes to the formation of collagen, and helps to better absorb Ca [63]. On the other hand, arginine is considered a conditionally essential amino acid [28]. Arginine can have an antithrombotic effect [64,65]. Hence, consuming deer meat rich in this amino acid may reduce the risk of vascular diseases. Methionine is involved in protein synthesis and the DNA methylation reaction. Thanks to its antioxidant properties, it also contributes to the maintenance of oxidative balance [66]. A recent study revealed its involvement in the regulation of the immune response [67]. In addition, methionine is essential in preventing vitamin B12 deficiency [66]. The SM muscles of the OFG deer and the CFG deer under analysis showed a high content of this amino acid, making this meat an attractive option for health-conscious consumers. Threonine is another amino acid present in large quantities in deer meat, which has a positive effect on the cardiovascular and immune systems, as well as on the condition of the liver. This amino acid strengthens the connective tissue and the muscles [29].

Tryptophan is classified as an LNAA amino acid [68] and is involved in the synthesis of vitamin PP [29]. It serves an important role in the regulation of mood and the adaptive response to stress [57]. Trp is essential for the cognitive, emotional, and energy functions of humans [68]. Deficiency of this amino acid can lead to serious diseases, e.g., diabetes mellitus, tuberculosis, cancer, and pellagra [25,69]. The recommended daily Trp intake in the diet for adults ranges from 250 to 425 mg/day [70]. It is sufficient to consume 50 g of deer meat in order to satisfy the demand for this amino acid. Deer meat is an excellent source of the most important amino acids in human nutrition. Amino acids such as methionine + cysteine, phenylalanine + tyrosine, and lysine are extremely important as they can help compensate for the loss of protein mass, whereas their deficiency in the diet limits the growth of children [44,71].

In addition, free amino acids play an important role in the flavor of foods. Alanine, glycine, proline, serine, and threonine are sweet in taste, whereas arginine, valine, tyrosine, tryptophan, phenylalanine, leucine, and isoleucine are characterized by a bitter taste [24]. Lysine was found to be one of the predominant amino acids in the muscles of deer from the forest feeding ground, and it is worth noting that it contributes to the salty flavor of the meat [72]. Six amino acids (Gly, Ile, Pro, Ser, Ala, and Glu) are directly linked to the umami taste, with Glu being the most important umami amino acid [73]. It follows from the authors’ own study that the Glu, Ser, and Ala contents are higher in the meat of the FFG deer and the CFG deer, which may contribute to the better taste of this meat than that of the OFG deer.

Accurate assessment of the nutritional quality of dietary proteins is of fundamental importance [74]. For the assessment of the nutritional value, the absorbability of the individual components by the human body is crucial. According to Leydig’s law, a particular amino acid is fully absorbable if it is present in protein in the appropriate amount. This is because the absorbability of amino acids in protein is determined by the exogenous amino acid that is present in the smallest range in relation to the required ratios. One of the methods used to check whether particular proteins in foods complement each other in terms of amino acid content is the limiting amino acid index (CS) [25]. The present study found that the muscles of the OFG, FFG, and CFG deer supply all the exogenous amino acids that are necessary to ensure the synthesis of protein from the meat consumed by humans in amounts that are considerably higher than the required minimum provided by the FAO/WHO benchmark [42,44]. Considering the standard (reference) protein recommended by FAO [45], it was found that the CS3 index for Lys reached the highest value in the present study (365.80–404.99%). Considerably lower CS indices for the maral (deer) meat were noted by Okuskhanova et al. [29]. All the samples analyzed in this study showed a complete profile of essential amino acids and a high-quality protein profile, which was higher than the standard [42,43,44]. The EAA content in relation to the standard protein (CS) ranges from 174.32 to 1354%. These results confirm those obtained previously for the muscles of the wild red deer by Lorenzo et al. [28], Perez-Serrano et al. [31], and Ugarković et al. [18]. 

The World Health Organization recommended that the total EAA and FAA contents should be sufficient to ensure ideal protein intake for infants, children, and adults and should be higher than 39%, 27%, and 11%, respectively [44]. Therefore, the ratio of the total EAA content to the total FAA content in all muscle samples, accounting for more than 44%, suggested that deer meat ensured favorable AA profiles and satisfactory nutritional values.

The PER index value exceeding 2.7 (considered the standard casein value) is regarded as an excellent source of protein [62]. The meat of the deer analyzed in this study exceeded this value two-fold, which means that the muscles contain a high-quality protein that provides an EAA composition that is complete and optimal for human needs. Therefore, as the quality of a protein increases, the required dietary protein level in the human diet decreases [75]. The calculated PER indices confirm the high nutritional value of the deer meat protein, which is also extremely important from a health-related perspective. This meat can be recommended for athletes who require higher protein intake to prevent considerable losses of lean muscle tissue, and for the elderly, it can help delay sarcopenia [76]. Intake of this protein increases bone mineral density [62]. In addition, it is recommended for those suffering from liver diseases and alcoholics [62]. In view of the lack of sufficient data on the nutritional value of deer muscle proteins in the literature, it is difficult to discuss this topic. PER values similar to those obtained in the present study were noted by Kowalska-Góralska [77] for *Acipenseridae* and *Salmonidae* fish eggs. According to Adeyeye et al. [75], who analyzed kilishi beef, the PER value was lower than that in the authors’ own study and ranged from 2.52 to 2.70.

## 5. Conclusions

This is the first study to compare the contents of particular amino acids and assess the quality of the protein in the muscles of the deer from the organic, conventional, and forest feeding grounds to provide clear evidence of significant differences in their composition. This study noted a higher exogenous and endogenous amino acid content in the LL and SM muscles of the deer from the forest feeding ground (FFG) and the conventional feeding ground (CFG) compared to the deer from the organic feeding ground (OFG). The differences in amino acids between the deer under assessment are probably due to the varied vegetation available to animals. The protein of OFG deer muscles, compared to that of CFG deer and FFG deer, showed the most balanced composition of essential to non-essential amino acids, which provides good potential for developing organic deer farming. Differences were also noted in the lysine content between the muscles of deer from the forest feeding ground and deer from the two other groups, which may contribute to the salty taste of that meat. In addition, the muscles of FFG deer and CFG deer exhibited a higher percentage content of both acidic and tasty amino acids. In contrast, the muscles of OFG deer were characterized by a higher percentage of branched-chain amino acids and large neutral amino acids compared to the muscles of CFG deer. The calculated expected PER and BV indices confirm the high nutritional value of the protein in the muscles of deer from the three feeding grounds. The nutritional value of deer muscle protein is extremely important for conscious consumers and should be displayed by meat breeders, producers, and retailers. The data presented in this study can be used professionally by nutrition specialists in preventive and therapeutic diets, as well as for scientific and research purposes. Further research should be conducted to better understand the differences observed between the deer having different diets to choose from and to learn more about the metabolic processes taking place in the body and the ability to synthesize and store amino acids in the muscles.

## Figures and Tables

**Table 1 animals-14-02763-t001:** The schema of the experiment.

Specification	Forest Feeding Grounds	Organic Feeding Grounds	Conventional Feeding Grounds
No of animals	12	12	12
Muscles	LL	LL	LL
	SM	SM	SM
Sex	♀ *n* = 6	♀ *n* = 6	♀ *n* = 6
	♂ *n* = 6	♂ *n* = 6	♂ *n* = 6

LL = longissimus lumborum; SM = musculus semimembranosus.

**Table 2 animals-14-02763-t002:** Standard protein amino acid profile.

Amino Acid	Standard FAO/WHO, 1991 [g/100 g Protein]	Standard USA, 2002 [g/100 g Protein]	Standard FAO, 2013 [g/100 g Protein]
Isoleucine	2.80	2.50	3.00
Leucine	6.59	5.50	6.10
Lysine	5.79	5.10	4.80
Methionine + cysteine	2.50	2.50	2.30
Phenylalanine + tyrosine	6.29	4.70	4.10
Threonine	3.39	2.70	2.50
Tryptophan	1.10	0.70	0.66
Valine	3.49	3.20	4.00
Histidine	-	-	1.60

FAO/WHO, 1991 [42]; USA, 2002 [43]; FAO, 2013 [44].

**Table 3 animals-14-02763-t003:** The protein (%) and amino acid content (mg/g of fresh tissue) of red deer *longissimus lumborum*.

Item	*Longissimus Lumborum* (LL)						
	FFG		OFG		CFG		
	N = 12		N = 12		N = 12		*p*-Value
	X	SE	X	SE	X	SE	
Protein	21.42 ^a^	0.66	23.29 ^b^	0.81	23.12 ^b^	0.78	0.001
Threonine (Thr)	9.23 ^b^	0.14	8.21 ^a^	0.22	8.61 ^a^	0.31	0.007
Valine (Val)	9.27 ^b^	0.75	8.53 ^a^	0.16	8.62 ^a^	0.28	0.011
Methionine (Met)	3.69 ^a^	0.16	4.06 ^a^	0.18	4.99 ^b^	0.16	<0.001
Isoleucine (Ile)	7.84 ^b^	0.15	7.08 ^a^	0.11	7.33 ^a^	0.22	0.021
Leucine (Leu)	14.88 ^b^	0.34	13.86 ^a^	0.32	15.05 ^b^	0.34	0.050
Phenyloalanine (Phe)	7.73 ^b^	0.15	6.95 ^a^	0.22	7.26 ^a^	0.21	0.028
Lysine (Lys)	18.02 ^b^	0.25	16.45 ^a^	0.30	16.84 ^a^	0.38	0.003
Tryptophan (Trp)	7.74	0.08	7.95	0.51	8.24	1.02	0.844
Histidine (His)	7.15 ^a^	0.15	7.07 ^a^	0.22	8.17 ^b^	0.31	0.003
Arginine (Arg)	11.89	0.13	11.80	0.17	11.25	0.33	0.113
Serine (Ser)	7.25 ^b^	0.11	6.66 ^a^	0.19	7.17 ^b^	0.27	0.035
Aspartic acid (Asp)	17.78 ^c^	0.12	15.71 ^a^	0.44	16.85 ^b^	0.56	0.005
Glutamic acid (Glu)	32.87 ^b^	0.38	26.71 ^a^	0.11	33.87 ^b^	0.84	<0.001
Proline (Pro)	8.09	0.15	7.87	0.23	7.80	0.11	0.355
Glycine (Gly)	7.85	0.18	7.47	0.28	7.63	0.27	0.457
Alanine (Ala)	10.75 ^b^	0.20	9.57 ^a^	0.26	10.39 ^b^	0.32	0.004
Cysteine (Cys)	0.61	0.05	0.59	0.02	0.58	0.02	0.806
Tyrosine (Tyr)	6.54 ^b^	0.06	6.08 ^a^	0.21	6.68 ^b^	0.21	0.012
TAA	194.11 ^b^	2.63	178.23 ^a^	4.03	195.33 ^b^	5.17	0.006
EAA	85.85 ^b^	1.25	81.40 ^a^	2.13	87.59 ^b^	2.81	0.045
NEAA	108.25 ^b^	1.78	96.82 ^a^	3.31	107.74 ^b^	3.80	0.003
EAA (%)	44.23 ^a^	0.26	45.78 ^b^	0.58	44.86 ^a^	0.59	0.048
NEAA (%)	55.76 ^c^	0.26	54.21 ^a^	0.58	55.13 ^b^	0.59	0.048
EAA:NEAA	0.79 ^a^	0.02	0.84 ^b^	0.02	0.81 ^a^	0.02	0.039
Met + Cys	4.50 ^a^	0.83	5.54 ^b^	0.61	6.01 ^b^	0.56	<0.001
Phe + Tyr	15.10 ^c^	0.32	13.45 ^a^	0.36	14.43 ^b^	0.42	0.025
EAA:TAA	0.44	0.08	0.45	0.16	0.44	0.05	0.950
Met:TAA	0.021	0.004	0.026	0.005	0.025	0.002	0.745
AAk	52.36 ^b^	0.81	44.51 ^a^	2.20	53.98 ^b^	2.11	<0.001
AAa	21.13	0.20	19.64	0.38	22.38	1.20	0.411
AAk (%)	26.97 ^b^	0.28	24.80 ^a^	0.78	27.60 ^b^	0.52	<0.001
AAa (%)	10.89	0.29	11.04	0.39	11.46	0.66	0.496
Met + Cys (%)	2.31 ^a^	0.17	3.08 ^b^	0.26	3.10 ^b^	0.27	<0.001
BCAA	32.38	0.52	30.74	0.63	32.34	0.95	0.053
BCAA (%)	16.68 ^a^	0.25	17.29 ^b^	0.67	16.52 ^a^	0.28	0.049
LNAA	47.48 ^b^	0.63	44.20 ^a^	0.94	46.82 ^ab^	1.41	0.035
LNAA (%)	24.46 ^ab^	0.22	24.85 ^b^	0.27	23.94 ^a^	0.24	0.018
DAA	72.07 ^b^	0.92	62.09 ^a^	3.05	72.72 ^b^	2.22	0.001
DAA (%)	37.13 ^b^	0.25	34.67 ^a^	0.64	37.21 ^b^	0.51	0.001
BCAA:AAa	1.44	0.02	1.41	0.03	1.42	0.06	0.849
EAAI1 (%)	168.79	2.51	164.75	3.23	180.25	6.07	0.353
EAAI2 (%)	207.97	2.65	202.99	3.48	224.43	8.26	0.499
EAAI3 (%)	228.30	2.71	222.84	4.03	247.13	10.20	0.522

^a, b, c^ Values within a row with different superscripts differ significantly at *p* < 0.05. FFG = forest feeding ground; OFG = organic feeding ground; CFG = conventional feeding ground; TAA = total amino acid; EAA = essential amino acids; NEAA = non-essential amino acids; AAk = acidic amino acids; AAa = aromatic amino acids; BCAA = branched-chain amino acids; LNAA = large neutral amino acids; DAA = tasty amino acids; EAAI1; EAAI2 and EAAI3 = essential amino acid index and amino acid standards for an adult from 1991, 2002, and 2013, respectively.

**Table 4 animals-14-02763-t004:** The protein (%) and amino acid content (mg/g of fresh tissue) of red deer *musculus semimembranosus*.

Item	*Musculus Semimembranosus* (SM)						
	FFG		OFG		CFG		*p*-Value
	X	SE	X	SE	X	SE	
Protein	22.25 ^a^	0.51	23.80 ^b^	0.64	23.34 ^b^	0.71	0.001
Threonine (Thr)	9.51 ^b^	0.16	8.54 ^a^	0.27	9.21 ^b^	0.32	0.017
Valine (Val)	9.54	0.27	8.82	0.19	9.00	0.30	0.074
Methionine (Met)	3.86 ^a^	0.23	4.52 ^b^	0.32	5.48 ^c^	0.51	0.017
Isoleucine (Ile)	7.77	0.19	7.43	0.11	7.63	0.32	0.466
Leucine (Leu)	15.08 ^ab^	0.25	14.50 ^a^	0.31	15.71 ^b^	0.48	0.049
Phenyloalanine (Phe)	8.19 ^b^	0.13	7.13 ^a^	0.23	7.60 ^ab^	0.27	0.001
Lysine (Lys)	18.21 ^b^	0.36	17.01 ^a^	0.29	17.33 ^a^	0.54	0.045
Tryptophan (Trp)	6.04	0.01	6.19	0.17	7.83	1.16	0.113
Histidine (His)	7.67	0.16	7.28	0.22	7.80	0.28	0.183
Arginine (Arg)	12.83	0.14	12.36	0.20	11.88	0.41	0.062
Serine (Ser)	7.42 ^b^	0.10	6.95 ^a^	0.16	7.58 ^b^	0.30	0.042
Aspartic acid (Asp)	18.37 ^c^	0.28	16.33 ^a^	0.38	17.64 ^b^	0.65	0.010
Glutamic acid (Glu)	34.00 ^b^	0.61	28.19 ^a^	0.95	36.29 ^b^	1.06	0.001
Proline (Pro)	8.39	1.16	8.08	0.27	8.14	0.21	0.715
Glycine (Gly)	8.68 ^b^	0.15	7.61 ^a^	0.23	8.11 ^ab^	0.35	0.007
Alanine (Ala)	11.03 ^b^	0.12	9.97 ^a^	0.22	10.69 ^ab^	0.28	0.006
Cysteine (Cys)	0.64	0.02	1.02	0.23	0.53	0.02	0.088
Tyrosine (Tyr)	6.91 ^b^	0.11	6.33 ^a^	0.11	6.89 ^b^	0.22	0.017
TAA	189.02 ^b^	2.01	172.60 ^a^	4.52	187.34 ^b^	4.60	0.006
EAA	85.33 ^b^	1.17	80.15 ^a^	1.92	85.11 ^b^	2.96	0.022
NEAA	103.68 ^b^	1.17	92.45 ^a^	330	102.23 ^b^	2.67	0.006
EAA (%)	45.14 ^a^	0.26	46.53 ^b^	0.51	45.41 ^a^	0.56	0.040
NEAA (%)	54.85 ^b^	0.26	53.46 ^a^	0.51	54.58 ^b^	0.56	0.045
EAA:NEAA	0.82 ^a^	0.05	0.87 ^b^	0.02	0.83 ^a^	0.01	0.043
Met + Cys	4.35	0.20	4.64	0.15	5.57	0.28	0.073
Phe + Tyr	14.27 ^b^	0.21	13.03 ^a^	0.31	13.94 ^b^	0.40	0.004
EAA:TAA	0.45	0.002	0.46	0.05	0.45	0.03	0.756
Met:TAA	0.01	0.001	0.02	0.001	0.02	0.001	0.711
AAk	50.65 ^b^	3.11	42.41 ^a^	2.51	50.72 ^b^	1.05	0.001
AAa	22.00 ^b^	0.12	20.98 ^a^	0.54	22.17 ^b^	1.11	0.025
AAk (%)	26.80 ^b^	0.10	24.41 ^a^	0.71	27.09 ^b^	0.42	0.001
AAa (%)	11.64	0.10	12.19	0.29	11.83	0.41	0.492
Met + Cys (%)	2.30 ^a^	0.10	2.70 ^b^	0.29	2.97 ^c^	0.29	0.023
BCAA	31.78	0.54	29.46	0.62	31.00	0.85	0.202
BCAA (%)	16.81 ^ab^	0.19	17.09 ^b^	0.16	16.53 ^a^	0.16	0.020
LNAA	46.05	0.52	42.50	0.85	44.94	0.91	0.064
LNAA (%)	24.36 ^ab^	0.14	24.64 ^b^	0.17	23.97 ^a^	0.18	0.020
Trp:LNAA	0.16	0.01	0.18	0.01	0.18	0.02	0.149
LNAA:Trp	5.96	0.13	5.75	0.65	6.53	0.86	0.640
DAA	69.25 ^b^	0.58	59.45 ^a^	2.92	68.74 ^b^	2.06	0.001
DAA (%)	36.63 ^b^	0.06	34.28 ^a^	0.72	36.70 ^b^	0.34	0.001
DAA:TAA	0.36 ^b^	0.02	0.34 ^a^	0.07	0.36 ^b^	0.13	0.001
BCAA:AAa	1.53	0.02	1.57	0.02	1.48	0.06	0.320
EAAI1 (%)	176.08 ^b^	1.86	169.13 ^a^	3.05	177.17 ^b^	6.74	0.029
EAAI2(%)	220.11 ^b^	1.13	212.36 ^a^	4.16	221.94 ^b^	9.55	0.034
EAAI3 (%)	242.17 ^b^	3.40	234.03 ^a^	4.86	244.65 ^b^	11.06	0.034

^a, b, c^ Values within a row with different superscripts differ significantly at *p* < 0.05 FFG = forest feeding ground; OFG = organic feeding ground; CFG = conventional feeding ground; TAA = total amino acid; EAA = essential amino acids; NEAA = non-essential amino acids; AAk = acidic amino acids; AAa = aromatic amino acids; BCAA = branched-chain amino acids; LNAA = large neutral amino acids; DAA = tasty amino acids; EAAI1; EAAI2 and EAAI3 = essential amino acid index and amino acid standards for an adult from 1991, 2002, and 2013, respectively.

**Table 5 animals-14-02763-t005:** Chemical index, protein efficiency ratio, and the biological value of the *longissimus lumborum*.

Item	*Longissimus Lumborum* (LL)						
	FFG		OFG		CFG		*p*-Value
	X	SE	X	SE	X	SE	
CS 1 Ile	277.44 ^b^	5.26	265.32 ^a^	5.05	272.56 ^ab^	9.24	0.021
CS 2 Ile	311.23 ^b^	6.05	297.64 ^a^	6.256	305.75 ^ab^	10.90	0.021
CS 3 Ile	258.25 ^b^	5.04	246.98 ^a^	5.80	253.71 ^ab^	9.61	0.021
CS 1 Leu	228.68	3.11	219.93	4.98	238.29	7.10	0.056
CS 2 Leu	273.89	3.82	263.41	6.13	285.39	9.06	0.056
CS 3 Leu	284.64	3.88	273.76	6.88	296.60	9.24	0.056
CS 1 Lys	314.37 ^b^	5.23	293.59 ^a^	5.44	299.20 ^a^	8.52	0.003
CS 2 Lys	356.74 ^b^	5.91	333.17 ^a^	6.54	339.53 ^a^	9.18	0.003
CS 3 Lys	404.99 ^b^	6.63	378.22 ^a^	6.84	385.45 ^a^	9.26	0.003
CS 1 Met + Cys	180.28 ^a^	10.32	221.95 ^b^	21.32	240.94 ^c^	20.84	<0.001
CS 2 Met + Cys	180.28 ^a^	10.32	221.95 ^b^	21.32	240.94 ^c^	20.84	<0.001
CS 3 Met + Cys	203.80 ^a^	11.58	250.91 ^b^	27.40	272.37 ^c^	21.21	<0.001
CS 1 Phe + Tyr	240.14 ^c^	3.84	214.00 ^a^	5.41	230.33 ^b^	7.10	0.025
CS 2 Phe + Tyr	321.00 ^c^	4.84	286.06 ^a^	6.93	307.89 ^b^	8.89	0.025
CS 3 Phe + Tyr	396.53 ^c^	6.00	353.37 ^a^	8.09	380.34 ^b^	11.15	0.025
CS 1 Thr	280.31 ^c^	3.01	251.69 ^a^	6.87	271.64 ^b^	10.14	0.007
CS 2 Thr	351.64 ^c^	4.02	315.73 ^a^	8.92	340.76 ^b^	13.83	0.007
CS 3 Thr	412.68 ^c^	5.06	370.55 ^a^	9.25	399.92 ^b^	15.27	0.007
CS 1 Trp	546.72	8.04	560.61	12.62	708.78	10.17	0.844
CS 2 Trp	857.36	12.06	879.14	21.91	1111.56	15.13	0.844
CS 3 Trp	992.73	14.51	1017.95	24.46	1287.00	18.83	0.844
CS 1 Val	273.58 ^b^	6.05	252.77 ^a^	6.02	258.12 ^a^	8.85	0.011
CS 2 Val	298.20 ^b^	6.48	275.52 ^a^	5.55	281.35 ^a^	9.45	0.011
CS 3 Val	317.66 ^b^	6.52	293.49 ^a^	6.76	299.71 ^a^	8.50	0.011
PER1	5.65	0.91	5.45	0.15	5.94	0.18	0.074
PER2	6.71 ^a^	0.21	7.02 ^a^	0.27	7.96 ^b^	0.24	0.018
PER3	5.72 ^b^	0.09	5.38 ^a^	0.11	5.70 ^b^	0.17	0.032
BV1	180.23	1.21	172.66	3.95	181.42	7.91	0.353
BV2	228.22	1.51	219.78	4.84	230.22	10.83	0.499
BV3	252.28	1.81	243.40	5.94	254.97	12.54	0.522

^a, b, c^ Values within a row with different superscripts differ significantly at *p* < 0.05. FFG = forest feeding ground; OFG = organic feeding ground; CFG = conventional feeding ground. CS 1 = chemical score and amino acid standards for an adult from 1991. CS 2 = chemical score and amino acid standards for an adult from 2002. CS 3 = chemical score and amino acid standards for an adult from 2013. PER1 = −0.468 + 0.454 × Leu − 0.104 × Tyr. PER2= −1.816 + 0.435 × Met + 0.780 × Leu + 0.211 × His − 0.944 × Tyr. PER3 = 0.08084 × (X7) − 0.1094, where X7 = Thr + Val + Met + Ile + Leu + Phe + Lys. BV = biological value.

**Table 6 animals-14-02763-t006:** Chemical index, protein efficiency ratio, and the biological value of the *musculus semimembranosus*.

Item	*Musculus Semimembranosus* (SM)						
	FFG		OFG		CFG		*p*-Value
	X	SE	X	SE	X	SE	
CS 1 Ile	279.88	5.92	252.70	5.10	261.93	8.03	0.466
CS 2 Ile	313.96	6.23	283.48	6.30	293.83	9.68	0.466
CS 3 Ile	260.52	5.61	235.23	4.85	243.82	8.95	0.466
CS 1 Leu	222.61	5.40	210.20	5.40	228.37	6.35	0.059
CS 2 Leu	266.62	5.64	251.75	5.85	273.51	7.38	0.059
CS 3 Leu	277.09	6.41	261.64	6.66	284.25	7.33	0.059
CS 1 Lys	311.06 ^c^	4.84	283.95 ^a^	5.60	290.67 ^b^	6.86	0.045
CS 2 Lys	352.99 ^c^	4.85	322.23 ^a^	5.98	329.85 ^b^	7.94	0.045
CS 3 Lys	400.72 ^c^	512	365.80 ^a^	6.68	374.46 ^b^	8.45	0.045
CS 1 Met + Cys	174.32	7.28	186.27	5.34	223.20	8.45	0.073
CS 2 Met + Cys	174.32	7.28	186.27	5.34	223.20	8.45	0.073
CS 3 Met + Cys	197.06	8.45	210.57	6.73	252.32	9.03	0.073
CS 1 Phe + Tyr	226.96 ^b^	3.24	207.28 ^a^	5.06	221.73 ^b^	5.01	0.004
CS 2 Phe + Tyr	303.39 ^b^	3.70	277.08 ^a^	7.81	296.39 ^b^	7.75	0.004
CS 3 Phe + Tyr	374.78 ^c^	4.92	342.28 ^a^	9.12	366.13 ^b^	9.05	0.004
CS 1 Thr	272.11 ^c^	3.54	242.01 ^a^	6.07	253.85 ^b^	8.83	0.017
CS 2 Thr	341.34 ^c^	4.73	303.59 ^a^	8.44	318.44 ^b^	10.16	0.017
CS 3 Thr	400.60 ^c^	5.46	356.30 ^a^	9.03	373.73 ^b^	13.44	0.017
CS 1 Trp	700.86	7.32	719.88	14.00	745.92	12.00	0.113
CS 2 Trp	1099.07	11.14	1128.96	61.96	1169.74	121.48	0.113
CS 3 Trp	1272.61	14.16	1307.15	72.11	1354.44	154.03	0.113
CS 1 Val	265.76	2.04	244.67	4.01	247.03	7.03	0.074
CS 2 Val	289.68	2.68	266.69	4.45	269.27	8.29	0.074
CS 3 Val	308.58	2.18	284.09	5.59	286.84	8.07	0.074
PER1	5.51	0.18	5.19	0.17	5.67	0.15	0.071
PER2	6.57 ^a^	0.31	6.50 ^a^	0.22	7.51 ^b^	0.31	0.004
PER3	5.58	0.07	5.15	0.11	5.44	0.14	0.074
BV1	172.29 ^a^	2.74	167.88 ^a^	3.45	184.78 ^b^	6.90	0.029
BV2	215.00 ^a^	2.37	209.57 ^a^	3.40	232.93 ^b^	9.72	0.034
BV3	237.15 ^a^	2.93	231.20 ^a^	4.82	257.67 ^b^	12.68	0.034

^a, b, c^ Values within a row with different superscripts differ significantly at *p* < 0.05. FFG = forest feeding ground; OFG = organic feeding ground; CFG = conventional feeding ground. CS 1 = chemical score and amino acid standards for an adult from 1991. CS 2 = chemical score and amino acid standards for an adult from 2002. CS 3 = chemical score and amino acid standards for an adult from 2013. PER1 = −0.468 + 0.454 × Leu − 0.104 × Tyr. PER2= −1.816 + 0.435 × Met + 0.780 × Leu + 0.211 × His − 0.944 × Tyr. PER3 = 0.08084 × (X7) − 0.1094, where X7 = Thr + Val + Met + Ile + Leu + Phe + Lys. BV = biological value.

## Data Availability

The data presented in this study are available on request from the corresponding author.

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
