# Peer review of "Amino Acid Content in the Muscles of the Red Deer (Cervus elaphus) from Three Types of Feeding Grounds"

_animals, 2024, doi:10.3390/ani14192763_

Round 1
Reviewer 1 Report
Comments and Suggestions for Authors
The study compared the amino acid content and evaluated the nutritional and biological value of deer muscle protein derived from three distinct feeding grounds: forest, organic, and conventional. Given the lack of substantial research in this area, the current work is novel and fills an important gap. Using appropriate methodology, the author has obtained a number of valuable data. The presentation of the results and their discussion do not raise any major questions.
Minor comments:
Since the work does not have line numbers I will refer to pages.
1. there are minor stylistic errors in the work. For example at page 1 double “of” can be found, please rephrase the vague sentences throughout the text.
2. please check carefully in the introduction and discussion when the reference is to meat (in general) and when to game meat (page 1, 2)
3. Page 3, nutritional value of what?
4. Page 3, chapter 2.1 – all samples were taken from animals shot at the same time?
5. Page 3, How the authors was able to precisly calculate the exact age of the aniamls. In other words: “exactly or c.a. 31 months old”?
6. Page 13, Conclusion: Try to rephrase the conclusions to avoid abbreviations. Using full names makes this section more clearable and self-explanatory.
Author Response
I thank the reviewer for their helpful comments. I addressed all comments and made changes as requested where appropriate.
Reviewer
The study compared the amino acid content and evaluated the nutritional and biological value of deer muscle protein derived from three distinct feeding grounds: forest, organic, and conventional. Given the lack of substantial research in this area, the current work is novel and fills an important gap. Using appropriate methodology, the author has obtained a number of valuable data. The presentation of the results and their discussion do not raise any major questions.
Minor comments:
Since the work does not have line numbers I will refer to pages.
- there are minor stylistic errors in the work. For example at page 1 double “of” can be found, please rephrase the vague sentences throughout the text.
Author: The reviewer is correct. I checked and deleted it.
Reviewer
- please check carefully in the introduction and discussion when the reference is to meat (in general) and when to game meat (page 1, 2)
Author: Thank you for your suggestion. I checked it.
Reviewer
- Page 3, nutritional value of what?
Author: I added the missing word “of meat”
Reviewer
- Page 3, chapter 2.1 – all samples were taken from animals shot at the same time?
Author: I corrected it. I added it in the manuscript explanation “at 48 h post mortem”.
Reviewer
- Page 3, How the authors was able to precisely calculate the exact age of the animals. In other words: “exactly or c.a. 31 months old”?
Author: Corrected. “Animals were approximately three years old as estimated by tooth eruption (it is of ca. 31months)”.
Reviewer
- Page 13, Conclusion: Try to rephrase the conclusions to avoid abbreviations. Using full names makes this section more clearable and self-explanatory.
Author: Thank you for your comment. Abbreviation of amino acids was not an appropriate term to use, so its full form has been written.
Thank you for your sincere and careful review on my manuscript.
Best regards,
Author
Reviewer 2 Report
Comments and Suggestions for Authors The publication is very interesting. It contains an innovative element regarding research on the composition of deer meat. I suggest to improve Materials nad methods chapter. I put my notes in attached file.

Author Response
I want to thank reviewer for taking the time and efforts to review my manuscript and giving my a lot of valuable suggestion. The comments are valuable and very helpful for revising and improving my paper. I revised the manuscript and gave point-to-point feedbacks according to the reviewer suggestions to form current version of manuscript.
Reviewer
Suppliers
Author: Thank you for your suggestion. It has corrected.
Reviewer
should be specified that is "wild deer population"
Author: Thank you for your careful attention. The reviewer is correct in this statement. It has been corrected according to your suggestion.
Reviewer
should be specified that these numbers refers to meat posses during the hunting
Author: Suggestions of reviewer has been taken in to account.
Reviewer
Change the order in following description as it is in first sentence of that paragraph. So first forest then oragnic then conventional. now its mixed and hard to understand
Author: I sincerely thank the reviewer for careful reading. I clarified this in the revised version.
Reviewer
Was the soil fertilized on any of the farms? Did you check the soil chemical composition? Especially micro and macroelements?
Author: The soil was not fertilized on the farms. The chemical composition of the soil was not tested.
Reviewer
use abbreviation
Author: It has corrected.
Reviewer
what kind of salt licks? Just pure NaCl or with some additives?
Author: Constant accessibility to multi-ingredient salt licks (SOLSEL® Wild) was provided on both farms. Ingredients of the SOLSEL® Wild: 97% sodium chloride, 1.3% calcium carbonate; 38% sodium, 0.8% calcium, 0.1% magnesium; zinc 1071 mg/kg, manganese 1000 mg/kg, copper 1158 mg/kg, iron 503 mg/kg, iodine 102 mg/kg, selenium 25 mg/kg.
Reviewer
Has the rumen content been examined o determine the composition of the food?
Author: The rumen contents have not been tested to determine feed composition. This is a preliminary study. In another paper, which I am now preparing, I decided to make the rumen content been examined to determine the composition of the food.
Reviewer
in all farms in the same time? Was it outdoors or indoors?
Author: Red deers in the farms were shooted within ca. 30 days in the same month.
Reviewer
so after animal were shooted ther carcasses were transported? Where how far and how long?
Author: Animals were approximately three years old as estimated by tooth eruption. After shooting, animals were immediately bled out and then were transported (ca. 30 km; 1h) under refrigerated conditions an lorry into the game carcass handling unit where was eviscerated.
Reviewer
Bold
Author: It has corrected.
Thank you for your sincere and careful review on my manuscript.
Best regards,
Author

Reviewer 3 Report
Comments and Suggestions for Authors
Dear Author,
Your paper demonstrates a remarkable level of detail and scientific rigor. As this is the first time these values have been studied within the context of varying feeding regimes, it represents a significant contribution to the existing literature.
The introduction is both clear and concise, with the references being relevant and up-to-date. The materials and methods section is highly descriptive, and I would suggest that a graphical representation of the experimental plan be provided to enhance clarity. The discussion is thorough and aligns well with the results presented.
Considering that fatty acid values are highly influenced by the physiognomy of the animals, I would like to ask the author whether it would be relevant to include the average weight and fat percentage values for the different groups. If these metrics vary significantly, they might offer a different perspective on the discussion of the results.
Some remarks:
- Line 52: The phrase "amounted to 0.084 kg"—is this per capita?
- Statistical analysis: "differences were considered significant at P<0.01 and P<0.05"—Could you clarify the specific conditions under which each of these significance levels was applied?
Minor editing. I recommend a few changes only to improve readability (e.g. avoid repeated use of very short sentences)
Author Response
Thank you for your comments concerning my manuscript. The comments are valuable and very helpful for revising and improving my paper. I have carefully studied the comments and made corrections and other changes.
Reviewer
Dear Author,
Your paper demonstrates a remarkable level of detail and scientific rigor. As this is the first time these values have been studied within the context of varying feeding regimes, it represents a significant contribution to the existing literature.
The introduction is both clear and concise, with the references being relevant and up-to-date. The materials and methods section is highly descriptive, and I would suggest that a graphical representation of the experimental plan be provided to enhance clarity. The discussion is thorough and aligns well with the results presented.
Author:
Author: The schema of the experiment in Material and Methods the table was added as well as the numeration of all tables in the manuscript has been changed.
Table 1. The schema of the experiment.
|
Specification |
Forest feeding grounds |
Organic feeding grounds |
Conventional feeding grounds |
|
No of animals |
12 |
12 |
12 |
|
Muscles |
LL |
LL |
LL |
|
|
SM |
SM |
SM |
|
Sex |
♀ n=6 |
♀ n=6 |
♀ n=6 |
|
|
♂ n=6 |
♂ n=6 |
♂ n=6 |
LL = longissimus lumborum; SM = musculus semimembranosus
Reviewer
Considering that fatty acid values are highly influenced by the physiognomy of the animals, I would like to ask the author whether it would be relevant to include the average weight and fat percentage values for the different groups. If these metrics vary significantly, they might offer a different perspective on the discussion of the results.
Author: Thank you for this comment, however the fatty acid profile (as an aspect of the same experiment) was published before and it is impossible to repeat it in current paper. Also it does not seem appropriate to take into account the percentage of fat content. In the case of wild animals not body weight but carcass weight is typically registered.
Reviewer
Some remarks:
Line 52: The phrase "amounted to 0.084 kg"—is this per capita?
Author: It has been corrected according to your suggestion.
Reviewer
Statistical analysis: "differences were considered significant at P<0.01 and P<0.05"—Could you clarify the specific conditions under which each of these significance levels was applied?
Author: Thank you for your comment. The sentence was rephrased more precisely. The differences were considered significant if P was smaller than 0.05.
Reviewer
Comments on the Quality of English Language.
Minor editing. I recommend a few changes only to improve readability (e.g. avoid repeated use of very short sentences)
Author: Suggestions of reviewer has been taken in to account.
Thank you for your sincere and careful review on my manuscript.
Best regards,
Author
